# Micromagnetic Modeling of All Optical Switching of Ferromagnetic Thin Films: The Role of Inverse Faraday Effect and Magnetic Circular Dichroism

**Victor Raposo**[ID]**, Rodrigo Guedas, Felipe García-Sánchez**[ID]**, M. Auxiliadora Hernández, Marcelino Zazo and Eduardo Martínez ***

Departamento de Física Aplicada, University of Salamanca, 37008 Salamanca, Spain; victor@usal.es (V.R.); rodriguedas@usal.es (R.G.); fgs@usal.es (F.G.-S.); auximl@usal.es (M.A.H.); marcel@usal.es (M.Z.)
* Correspondence: edumartinez@usal.es



**Featured Application: Existing magnetic recording devices require the application of pulsed strong magnetic fields and/or electrical currents. The duration of these typical pulses needed to reverse the initial state at the nanoscale lies in the nanosecond regime, which imposes a limit for the maximum speed of writing operations. Manipulation of the magnetic state by ultrashort laser pulses is interesting for the future development of novel magnetic recording devices. Differently from conventional field or current pulses, the deterministically control of the magnetic state by means of ultrashort laser pulses with duration of a few femtoseconds will constitute a major step for next future of laser induced recording devices. The present manuscript provides novel and efficient numerical methods which allow us to explore the physics behind all-optical switching and domain wall motion observations in ferromagnetic systems with perpendicular anisotropy.**

**Abstract:** There is a lot of experimental evidence of All Optical Switching (AOS) by applying ultrashort laser pulses on ferromagnetic thin films with perpendicular magnetic anisotropy. However, the physical origin behind these processes remains under debate. In addition to the heating caused by the laser pulses, the Inverse Faraday Effect (IFE) and Magnetic Circular Dichroism (MCD) have been proposed as the most probable phenomena responsible for the observations of helicity-dependent AOS. Here, we review the influence of both phenomena by means of realistic micromagnetic simulations based on the Landau–Lifshitz–Bloch equation coupled to the heat transport caused by the laser heating. The analysis allows us to reveal the similarities and differences between both effects. While both mechanisms may lead to the local inversion of the initial magnetic state of a ferromagnetic sample submitted to a train of circularly polarized laser pulses, the Inverse Faraday Effect proves to be more efficient for nucleation and domain wall movement and it reproduces more accurately the different magnetic configurations that the experiments report for different values of the fluence of the laser beam.

**Keywords:** micromagnetism; helicity-dependent all optical switching; Landau-Lifshitz-Bloch equation; two temperatures model; Inverse Faraday Effect; magnetic circular dichroism

## 1. Introduction

Manipulation of magnetism using ultrafast laser pulses without any external magnetic field, also called All Optical Switching (AOS), is fundamentally interesting and promises for low-power and high-speed spintronic devices. After the pioneer observations of ultrafast demagnetization in a Nickel film submitted to a femtosecond laser pulse [1], other experiments were performed to

manipulate the magnetization with laser pulses. The first studies of AOS appeared in ferrimagnetic alloys [2,3], synthetic ferrimagnets [4,5], and more recently, in ferromagnetic materials [6–9], which have relevance for ultrafast magnetic recording applications. The optical control of the magnetic state in a ferromagnetic sample usually requires series of laser pulses with circular polarization. This multi-shot helicity-dependent control of the magnetization has been observed in three different types of experiments. (i) The local area under the laser beam of a ferromagnetic thin film with perpendicular magnetic anisotropy (PMA) can be reversed depending on the initial magnetic state and on the helicity of the laser pulses. This phenomenon is called Helicity-Dependent All-Optical Switching [3,10]. (ii) Trains of laser pulses have been also applied on a thin film initially containing a domain wall (DW) separating up and down magnetic states. In some of these experiments, the center of the laser beam is located away from the DW center, and multi-shot laser pulses promote a displacement of the DW position if the laser helicity is properly chosen. These processes are commonly referred to as Helicity-Dependent All-Optical DW Motion (HD-DWM) [8–11]. (iii) In other experimental setups, the laser beam is sweep along a given path over the ferromagnetic thin film. Similarly to the HD-AOS case for a fixed laser beam location on top of the ferromagnetic thin-film, the switching of the local magnetic state along the laser beam path can only be achieved for a given helicity of the laser pulses, which depends on the initial magnetic state [10].

The full comprehensive role of the ultrafast laser pulses in all these multi-shot helicity dependent processes is still under debate, and in general, there are several effects which need to be considered. The heating and eventual demagnetization of the sample at the femtosecond scale is a well established effect [12,13]. During the application of a laser pulse, the temperature of the sample can overcome the Curie threshold resulting in a local demagnetized state where the ferromagnetic order is lost. Linear and circular polarizations of the laser pulse heats the sample, but if we consider the MCD, there are different absorption rates for each helicities, typically differing in some units of percent. The helicity-dependent optical switching of the initial magnetic state or domain wall motion may be determined by the Inverse Faraday Effect (IFE) [3,14–16], the Magnetic Circular Dichroism (MCD) [17,18], or the laser-induced spin currents [19,20]. According to the IFE, an angular momentum density is associated to the circularly polarized laser pulse [21] due to the photon spins that leads to the induction of an out-of-plane effective field [22]. The direction of this magneto-optical effective field ($\vec{B}_{MO}$) is determined from the laser helicity ($\sigma$) [23], which favors a direction for the magnetic state [3]. On the contrary, if the laser beam is linearly polarized, no out-of-plane field is induced. In the framework of the MCD, it is assumed that the energy absorption of the magnetic sample from the laser beam depends on the laser helicity (right- or left-handed helicity, $\sigma^{\pm} = \pm 1$) and the local magnetic state (up $\vec{m} \uparrow$, or down $\vec{m} \downarrow$). For instance, it is expected that a region with down magnetization ($\vec{m} \downarrow$) submitted to a laser pulse with right-handed helicity ($\sigma^{+} = +1$) absorbs more energy than the same region with opposite magnetization ($\vec{m} \uparrow$), and vice versa for the opposite laser helicity ($\sigma^{-} = -1$). If the laser is linearly polarized ($\sigma^{0} = 0$), the absorption energy rate is the same for both up and down magnetic states. Therefore, a given helicity promotes the magnetization reversal whereas the other supports the initial state of the magnetization. Although some numerical studies have been performed to describe the role of the MCD in ferrimagnetic samples [17], no realistic numerical simulations based on MCD have been presented for ferromagnetic materials.

Some theoretical trials have been presented in the past trying to describe the switching of the magnetization in these AOS processes. Those approaches include atomistic simulations [24] or solving the magnetic three temperatures model (M3TM) coupled to a simplified dynamical equation [1,25,26]. However, a realistic evaluation of the role of the IFE and the MCD has not been carried yet. One of main reasons for the lack of such a realistic description is due to numerical limitations. Indeed, as the samples used for AOS are large, with sizes of hundreds of $\mu m^2$, solving the full micromagnetic problem is a complicated task. Moreover, the involved time scales in these processes also differ over a wide range, going from the femtosecond scale of the laser pulses, the picosecond scale of the temperature evolution, and nanosecond scale for the magnetization dynamics and temperature

dissipation to the substrate. Additionally, the temperature even exceeds the Curie temperature ($T_C$), and consequently, LLB equation must be numerically solved. This requires small time stepping, which makes the numerical problem even more time-consuming. To date, there has been no full micromagnetic simulator that takes into account all the physics involved in extended samples and consequently, a realistic description of available HD-AOS experiments is still missing. Therefore, it has not been possible to realistically elucidate the real role of the IFE and the MCD in helicity-dependent all-optical switching or domain wall motion processes, which is precisely the aim of the present work.

Here, we present a full micromagnetic formalism based on a model that couples the laser heating, described by the two temperatures model (2TM) [3,12], to the magnetization dynamics governed by the stochastic LLB Equation [12,27], which includes the transient magnetic field caused by the IFE and/or the different energy absorption rates due to the MCD. Our analysis reveals the scope of both effects in two different helicity-dependent experiments: switching and domain wall motion. Thus paper is structured into the following sections: in Section 2, we introduce the theoretical framework we have developed to model helicity-dependent all-optical processes. Section 3 presents our numerical results for the two mentioned problems. Section 3.1. analyzes the Helicity-Dependent All-Optical Switching (HD-AOS), where starting from a uniformly magnetic state, the switching by circular polarized laser pulses is evaluated. In Section 3.2, we studied the Helicity-Dependent Domain Wall Motion (HD-DWM). In both subsections, the role of the IFE or the MCD is studied in a separated manner, and the analysis allows us to check under which conditions these phenomena describe the experimental observations. The conclusions are exposed in Section 4.

## 2. Micromagnetic Model

In a typical AOS experiment, a ferromagnetic layer grown over a substrate is subjected to the action of trains of laser pulses (Figure 1a), with a typical duration ($\tau_L$) from hundreds of femtoseconds to several picoseconds. The laser spot is assumed to have a space Gaussian profile with a full-width at half-maximum (FWHM) defined by the diameter $d_0$. Its temporal profile is assumed to be Gaussian, with $\tau_L$ being its FWHM duration. Therefore, the space and temporal profile of the laser power is

$$P(r,t) \;=\; P_0 \exp\left[-\frac{r^2}{d_0^2/(4ln2)}\right]\exp\left[-\frac{(t-t_0)^2}{\tau_L^2/(4ln2)}\right] \tag{1}$$

where $r \;=\; \sqrt{x^2+y^2}$ is the distance from the center of the laser spot and $t_0$ is the time at which the laser power reaches its maximum power ($P_0$) in the center of the laser spot. The maximum power of the laser is $P_0 \;=\; F/(t_{FM}\tau_L)$, where $F$ is the laser fluence and $t_{FM}$ is the thickness of the ferromagnetic sample.

Laser pulses heat the ferromagnetic sample. The temperature evolution in the system under the action of the laser pulses is described by the two Temperature Model (2TM) in terms of two subsystems involving the electron ($T_e$) and the lattice ($T_l$) temperatures [28]. The temperature dynamics are given by the following coupled set of differential equations:

$$\begin{aligned}
C_e\frac{\partial T_e}{\partial t} &\;=\; -k_e\nabla^2 T_e - g_{el}(T_e - T_l) + \eta(\vec{m},\sigma)(1-R)P(\vec{r},t)\\
C_l\frac{\partial T_l}{\partial t} &\;=\; -k_l\nabla^2 T_l - g_{el}(T_l - T_e)
\end{aligned} \tag{2}$$

where $e$ and $l$ refer to electrons and lattice, respectively. $C_i$ is the thermal capacity (in $\left[\text{J}/\left(\text{m}^3\text{K}\right)\right]$) and $k_i$ is the thermal conductivity (in $[\text{W}/(\text{m·K})]$) of each subsystem ($i : e, l$). $g_{el}$ is the coupling constant between electrons and lattice (in $\left[\text{W}/\text{m}^3\text{K}\right]$). $P(\vec{r},t)$ is the laser power, which is absorbed by the sample in a magnitude that depends on reflectivity of the sample ($R$), the local magnetization and helicity ($\sigma$) when the MCD is considered through the coefficient $\eta(\vec{m},\sigma)$. Note that above the Debye temperature, $C_l$ can be considered as constant parameter, whereas, $C_e$ is linear with the electron's temperature, $C_e \;=\; \gamma_e T_e$ where $\gamma_e$ is the electronic heat capacity (in $\text{J}/\text{m}^3\text{K}^2$). The influence of the substrate is

taken into account by adding an additional Newton-like term to the right hand side of the second one of these equations, $(-C_l(T - T_{sub})/\tau_{sub})$, where $T_{sub}$ is the substrate temperature, and the $\tau_{sub}$ is a characteristic time describing the heat transport to the substrate and the surrounding.

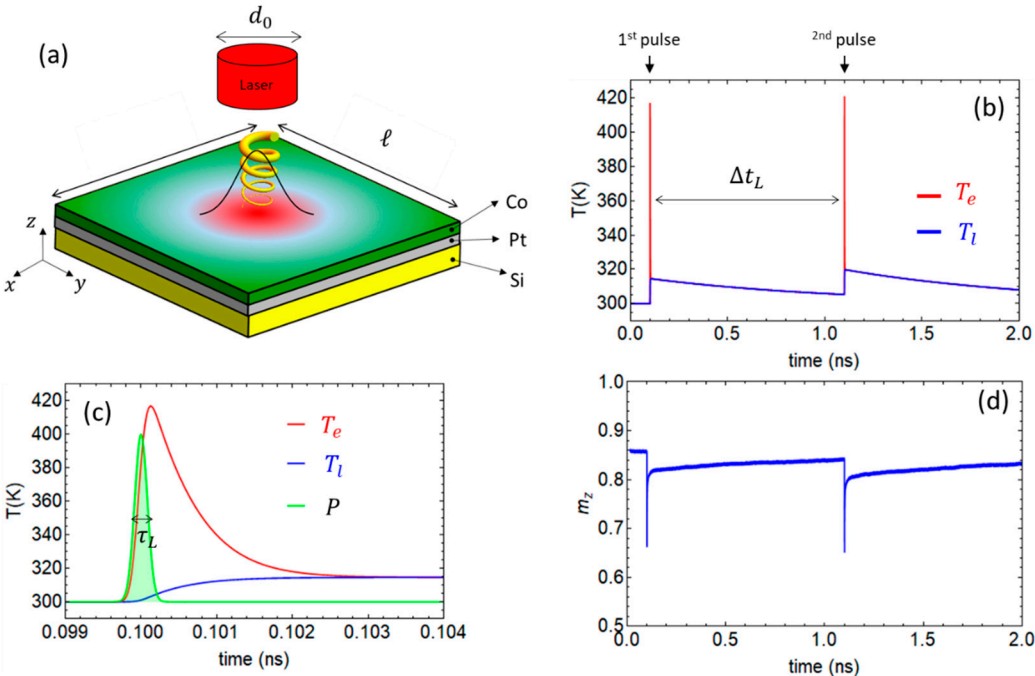

**Figure 1.** (**a**) Schematic of the Pt/Co thin film and the laser setup. (**b**) Time dependence of electron (red line) and lattice (blue line) temperatures obtained from the 2TM model for two laser pulses. (**c**) Detail of the temperature evolution during the time interval when the laser pulse is present, showing the differences between both electrons (red) and lattice (blue). The green line represents the normalized laser pulse power for comparison. (**d**) Temporal evolution of the averaged out-of-plane component of the magnetization ($m_z$) showing the ultrafast reduction due to the heating generated by the laser pulses.

As the laser pulse usually heats the sample close or even over its Curie temperature ($T_C$), the Landau–Lifshitz–Gilbert equation (LLG) cannot be used to describe the magnetization dynamics. Instead, the Landau–Lifshitz–Bloch (LLB) equation must be employed [27,29]. In this case, the dynamics of the normalized magnetization is given by the equation [29]:

$$\frac{d\vec{m}(\vec{r},t)}{dt} = -\gamma_0'\vec{m} \times \vec{H}_{eff} - \gamma_0'\frac{\alpha_\perp}{m^2}\left[\vec{m} \times \left(\vec{m} \times \left(\vec{H}_{eff} + \vec{H}_{th}^\perp\right)\right)\right] + \gamma_0'\frac{\alpha_\parallel}{m^2}\left(\vec{m} \cdot \vec{H}_{eff}\right)\vec{m} + \vec{H}_{th}^\parallel \quad (3)$$

where $\vec{m}(\vec{r},t) = \vec{M}(\vec{r},t)/M_s^0$ is the normalized magnetization with $M_s^0$ the saturation magnetization at $T = 0$ K. In the following we denote $m = m(T) \equiv |\vec{m}|$, as the magnitude of $\vec{m}$ depends on the local temperature because its equilibrium magnitude, $m_e(T)$, is temperature-dependent. $\gamma_0' = \gamma_0/(1 + \alpha^2)$ with $\gamma_0$ being the gyromagnetic ratio and $\alpha$ is the Gilbert damping. $\alpha_\parallel$ and $\alpha_\perp$ are the longitudinal and transverse damping parameters given by

$$\begin{aligned}\alpha_\parallel &= \alpha\left(\tfrac{2T}{3T_C}\right), \alpha_\perp = \alpha\left(1 - \tfrac{T}{3T_C}\right) \quad && T < T_c \\ \alpha_\parallel &= \alpha_\perp = \alpha\left(\tfrac{2T}{3T_C}\right) \quad && T > T_c\end{aligned} \quad (4)$$

The effective field $\vec{H}_{eff}$ in Equation (3) includes all the conventional micromagnetic interactions plus the magneto-optical field due to the IFE ($\vec{H}_{MO} = \vec{B}_{MO}/\mu_0$) that will be described later. Namely,

$$\vec{H}_{eff} = \vec{H}_{exch} + \vec{H}_{DM} + \vec{H}_{dmg} + \vec{H}_{ani} + \vec{H}_m + \vec{H}_{MO} \tag{5}$$

where $\vec{H}_{exch}$ is the exchange contribution, $\vec{H}_{DM}$ is the DMI interaction, $\vec{H}_{dmg}$ is the demagnetizing field, and $\vec{H}_{ani}$ is the magnetic anisotropy. Further details on these contributions of the effective field can be found in [30]. $\vec{H}_m$ represents the internal field in the LLB Equation (3), which is given by

$$\vec{H}_m = \begin{cases} \frac{1}{2\chi_{\parallel}}\left(1 - \frac{m^2}{m_e^2}\right)\vec{m}, & T < T_C \\[2mm] -\frac{1}{\chi_{\parallel}}\left(1 + \frac{3}{5}\frac{T_C m^2}{(T - T_C)}\right)\vec{m}, & T > T_C \end{cases} \tag{6}$$

where $\chi_{\parallel}$ is the longitudinal susceptibility,

$$\chi_{\parallel} = \left(\frac{\partial m_e}{\partial H_{ext}}\right)_{H_{ext}\to 0} = \frac{\frac{\mu_0 \mu_B}{k_B T}B'(x)}{\left(1 - \frac{T_C}{T}B'(x)\right)} \tag{7}$$

where $B'(x) = \frac{\partial B(x)}{\partial x}$ with $B(x)$ being the Brillouin function with $x = \frac{T_c}{T}m_e$ [31], $k_B$ the Boltzmann constant, and $\mu_B$ the Bohr magneton. $\vec{H}_m$ is responsible of the longitudinal relaxation of the magnetization towards its equilibrium value $m_e(T)$.

The LLB Equation (3) also includes stochastic terms $\vec{H}_{th}^{\perp}$ and $\vec{H}_{th}^{\parallel}$ to account for stochastic fluctuations due to the thermal noise [27]. The first one ($\vec{H}_{th}^{\perp}$) is a random thermal field orthogonal to the local magnetization, whereas the second one ($\vec{H}_{th}^{\parallel}$) describes the longitudinal noise, parallel to the local magnetization. Their statistical properties are summarized by

$$\begin{aligned} H_i^{\perp}(t) &= 0 \\ \overline{H_i^{\perp}(\vec{r},t)H_j^{\perp}(\vec{r}',t')} &= \frac{2K_B T(\alpha_{\perp} - \alpha_{\parallel})}{\gamma_0' M_s^0 V \alpha_{\perp}^2}\delta_{ij}\delta(t-t')\delta(\vec{r}-\vec{r}') \\ H_i^{\parallel}(t) &= 0 \\ \overline{H_i^{\parallel}(\vec{r},t)H_j^{\parallel}(\vec{r}',t')} &= \frac{2\gamma_0' T \alpha_{\parallel}}{M_s^0 V \alpha_{\perp}^2}\delta_{ij}\delta(t-t')\delta(\vec{r}-\vec{r}') \\ \overline{H_i^{\perp}(\vec{r},t)H_j^{\parallel}(\vec{r}',t')} &= 0 \end{aligned} \tag{8}$$

In these expressions, the notation $\overline{\cdots}$ indicates the average over different stochastic realizations of the noise, and the sub-indexes are the Cartesian components, $i, j : x, y, z$. $V$ is the volume of the computational cell (see [32] for details).

By solving Equations (2) and (3) we can obtain the dynamical behavior of the magnetization as a function of temperature in a self-consistent way. However, some other phenomena need to be considered when circularly polarized laser pulses are applied. IFE or MCD are taken into account in the micromagnetic formalism as follows:

(a) *Inverse Faraday Effect* (IFE). The generation of a magnetic field from a circularly polarized laser beam has been demonstrated experimentally [33] by measuring the amplitude of magnetic precession under different helicities. Theoretically, the IFE [34] predicts the existence of an optically induced magnetic field which can be described as

$$\vec{B}_{MO}(\vec{r},t) = \mu_0 \vec{H}_{MO}(\vec{r},t) = (\sigma^{\pm})B_{MO}^0 f_{MO}(\vec{r},t)\vec{u}_z \tag{9}$$

where $B^0_{MO}$ determines the maximum value of the $\vec{B}_{MO}(\vec{r}, t)$, which occurs at $t = t_0$ in the center of the laser spot. This field points along the out-of-plane direction ($\vec{u}_z$) and its sense, either positive or negative, is imposed by the helicity of the laser helicity: $\sigma^+ = +1$ for right-handed (RC), and $\sigma^- = -1$ for left-handed (LC) circular polarizations. The space-temporal dependence of the magneto-optical field is described by the function $f_{MO}(\vec{r}, t)$:

$$f_{MO}(\vec{r}, t) = \begin{cases} \exp\left[-\dfrac{r^2}{d_0^2/(4\ln 2)}\right]\exp\left[-\dfrac{(t-t_0)^2}{\tau_L^2/(4\ln 2)}\right], & t < t_0 \\[2em] \exp\left[-\dfrac{r^2}{d_0^2/(4\ln 2)}\right]\exp\left[-\dfrac{(t-t_0)^2}{(\tau_L+\tau_d)^2/(4\ln 2)}\right], & t > t_0 \end{cases} \tag{10}$$

with $\tau_d$ accounts for the possible delay of the magneto-optical field decay with respect to the power of laser pulse [3,35]. According to the theoretical description of the IFE, the magnitude $B^0_{MO}$ should be related to the laser fluence $F$ and that IFE susceptibility $\chi_{IFE}$. However, as it is difficult to infer $\chi_{IFE}$ from first principles or measure it from experimental means, in the present study, we assume that $F$ and $B^0_{MO}$ are independent parameters.

(b) *Magnetic circular Dichroism* (MCD). Some experiments suggest that the energy absorption rate of a PMA ferromagnetic sample depends on the its magnetic state (up $\vec{m}\uparrow$ or down $\vec{m}\downarrow$), and its energy absorption rate may be different for right (RC, $\sigma^+$) and left (LC, $\sigma^-$) circular polarizations [10]. To model the MCD we consider that the up state ($\vec{m}\uparrow$) absorbs more power from the laser pulse for the left-handed polarization than for the right-handed circular polarizations. The opposite happens for the down state ($\vec{m}\downarrow$). We characterize this different absorption rate by the MCD coefficient, defined as [17]

$$\text{MCD}(\%) = \frac{A_{LC} - A_{RC}}{A_{LP}} \times 100 \tag{11}$$

where $A_i$ with $i : LC, RC, LP$ are dimensionless parameters representing the absorption rate for left-handed circular ($\sigma^-$), right-handed circular ($\sigma^+$) and linear polarizations of the laser pulse respectively. MCD is included in Equation (2) by the coefficient $\eta(\vec{m}, \sigma)$, which accounts the different absorption rates depending on the laser polarization and the magnetic state. Typically, MCD is a function of frequency obtained by ellipsometry measurements that leads to MCD(%) values of some units [17,36]. In our micromagnetic model, as $\vec{m}(\vec{r}, t)$ is a continuous function of position and time, we assume a linear dependence with the out-of-plane component ($m_z$) as follows:

$$\eta(\vec{m}, \sigma) = A_{LP}[1 - \sigma^\pm m_z \, \text{MCD}(\%)] \tag{12}$$

where $\sigma^\pm = \pm 1$. The function $\eta(\vec{m}, \sigma)$ represents the variation of power absorption from the circularly polarized laser relative to the absorption for the linearly polarized laser pulse ($A_{LP}$). For example, a region with $m_z = +1$ has a minimum absorption for right-handed ($\sigma^+$) pulses ($\eta(+1, \sigma^+) = A_{LP}[1 - \text{MCD}(\%)]$), and a maximum absorption for left-handed ($\sigma^-$) pulses ($\eta(+1, \sigma^-) = A_{LP}[1 + \text{MCD}(\%)]$). The absorption rate is linear with $m_z$, which varies from $-1$ to $+1$.

In the present work, we simulate a Pt/Co bilayer on top of a Silicon wafer. A square-shape thin film Co layer of side $\ell$ and thickness $t_{FM}$ with high PMA is submitted to a train of laser pulses of full width at half power of $d_0 = \ell/2$. For the simulations, we adopted typical thermal and micromagnetic parameters for a Pt/Co bilayer. These inputs are collected in Table 1. $\ell$ was varied from 1.5 µm to 12 µm to check the dependence with the size of the laser beam. Disorder was taken into account in the form of grains, with a random a variation of the eas*y*-axis direction of the PMA of $\pm 3^0$ from grain to grain, being 15 nm the characteristic grain size [37].

**Table 1.** Micromagnetic and thermal parameters employed in the simulations.

| $M_S\left(\frac{\text{kA}}{\text{m}}\right)$ | $k_u\left(\frac{\text{MJ}}{\text{m}^3}\right)$ | $A_{ex}\left(\frac{\text{pJ}}{\text{m}}\right)$ | $D\left(\frac{mJ}{\text{m}^3}\right)$ | $K_e\left(\frac{\text{W}}{\text{m}\cdot\text{K}}\right)$ | $\gamma_e\left(\frac{\text{J}}{\text{m}^3\text{K}^2}\right)$ | $C_L\left(\frac{\text{MJ}}{\text{m}^3\text{K}}\right)$ | $g_{el}\left(\frac{\text{W}}{\text{m}^3\text{K}}\right)$ | $T_C(\text{K})$ | $\tau_{subs}(\text{ns})$ |
|---|---|---|---|---|---|---|---|---|---|
| 1.1 | 1.25 | 15 | 2.25 | 91 | 930 | 3.7 | $6\times10^{17}$ | 550 | 0.9 |

The Co layer was discretized using a finite difference scheme. The in-plane size of the computational cells is $\Delta x = \Delta y = 3$ nm, while $\Delta z = 0.8$ nm coincides with the thickness of the Co layer ($t_{FM}$). Micromagnetic simulations, solving Equations (2) and (3) simultaneously, were performed by using a Heun integration scheme with an adaptive time step: during each laser pulse, which is $\tau_L = 200$ fs long, the time step was set to $\Delta t = 1$ fs. Once each laser pulse is turned off, the time step was enlarged to $\Delta t = 25$ fs to speed up the computations. Several tests were performed with reduced cell sizes and time steps to ensure the numerical validity of the presented results.

## 3. Results

As it has already been mentioned, it has been experimentally proven that a train of circularly polarized laser pulses acting on a PMA ferromagnetic sample can lead to a magnetization switching [10] depending on the laser helicity ($\sigma^\pm$) and the initial magnetic state (up $\vec{m}\uparrow$ or down $\vec{m}\downarrow$). Before analyzing the role of IFE and MCD in these HD-AOS processes, we first describe the heating caused by the laser, which is common for both IFE and MCD scenarios.

The heating induced by the laser pulses is described by the 2TM, Equation (2). Figure 1b shows a typical temporal evolution of the temperature of the electrons ($T_e$) and lattice ($T_l$) in the center of the sample under two consecutive laser pulses, with $\Delta t_L = 1$ ns being the time between them. Due to the low electron thermal capacity, $T_e$ increases faster than $T_l$ (see Figure 1c). Electrons and phonons equilibrate their temperatures sometime after the laser pulse is switched off (see Figure 1b), and their temperatures relax towards the substrate temperature ($T_{sub} = 300$ K) for longer times. Therefore, the first effect of the laser is the sudden temperature increase of the electron temperature, which with high enough fluence can reach or even overcome $T_C$. In such a situation, the sample demagnetizes locally in the area exposed to the laser beam. An example of this demagnetization process is shown in Figure 1d, where the out-of-plane magnetization component averaged over the sample volume is plotted as function of time during the application of two laser pulses. In order to promote the switching of the local magnetization under the laser spot, additional ingredients that induce some asymmetry depending on the helicity of the laser are needed. In what follows, we will evaluate the effect of the IFE and the MCD in two different helicity-dependent all-optical processes: the local magnetization switching (Section 3.1) and the domain wall motion (Section 3.2).

### 3.1. Helicity-Dependent All Optical Switching (HD-AOS)

In the present section, we present numerical results trying to reproduce the experimental observation of the HD-AOS by evaluating two different scenarios: the IFE and the MCD, which we evaluated separately in Section 3.1.1 (IFE) and Section 3.1.2 (MCD). We start from a uniformly magnetized state (either up $\vec{m}\uparrow$ or down $\vec{m}\downarrow$), and a series of laser pulses with circular polarization (either LC or RC) are applied.

#### 3.1.1. Helicity-Dependent All Optical Switching with Inverse Faraday Effect

We firstly evaluate if the IFE can reproduce the experimental observation for the different combinations of helicity and initial state of the sample. Therefore, here, the MCD is not taken into account and the results are obtained by assuming that the IFE is the only helicity-dependent phenomenon. Figure 2 shows transient magnetization snapshots of a sample with $\ell = 12\ \mu\text{m}$ submitted to a train of 25 laser pulses with $\tau_L = 200$ fs. The IFE generates an effective magneto-optical field with $B_{MO,MAX} = 5$ T, which has a delay of $\tau_d = 400$ fs with respect to the laser power. The laser

fluence is $F = 0.55$ J/m$^2$. It can be seen that after 25 pulses only the laser with helicity that creates a transient magneto-optic field ($\vec{B}_{MO}$) pointing in the opposite direction to the initial state, produces the local switching of the magnetization. This happens for $(\vec{m}\uparrow, \sigma^-)$ (Figure 2b) and for $(\vec{m}\downarrow, \sigma^+)$ (Figure 2c). No local switching is achieved for the rest of combinations ($(\vec{m}\uparrow, \sigma^+)$ or $(\vec{m}\downarrow, \sigma^-)$), where the final state is the same as the initial one (see last snapshots in Figure 2a,d). These results are in accordance with those published in [8], where a very large number of laser pulses were required to promote the HD-AOS. According to these micromagnetic results, the IFE is revealed as a possible cause of the AOS, qualitatively reproducing the experimental observations.

Although the IFE scenario seems to be consistent with observations, the magnitude of the magneto-optical effective field ($B_{MO}^0$) or its delay with respect to the laser pulse ($\tau_d$) are difficult (if possible) to determine by experimental means. However, our numerical formalism allows us to evaluate the magnetization dynamics under laser pulses with both $B_{MO}^0$ and $\tau_d$ as being free input parameters. As the main uncertainty is in the value of $B_{MO}^0$, which is essentially unknown in experimental setups, we have studied the HD-AOS dependence on $F$ and $B_{MO}^0$ under series of laser pulses with fixed delay ($\tau_d = 400$ $fs$). The micromagnetic results are summarized in Figure 3, where the initial state is up ($\vec{m}\uparrow$) and the helicity is left-handed ($\sigma^-$), the one which generates a field $\vec{B}_{MO} \propto -\vec{u}_z$ pointing in the opposite direction to the initial magnetic state, and consequently promotes the switching depending on $F$ and $B_{MO}^0$.

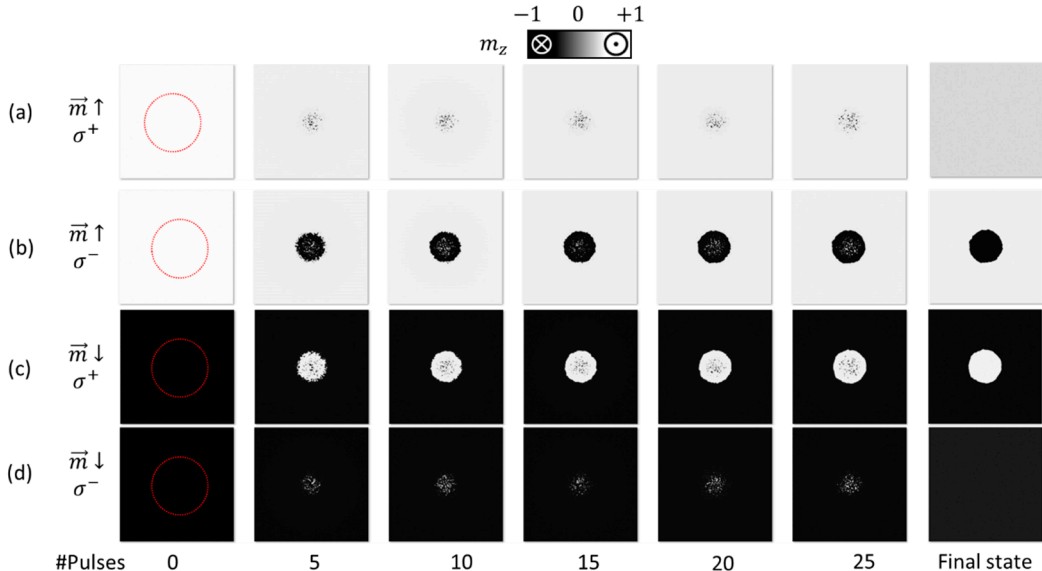

**Figure 2.** Snapshots of the magnetization after different laser pulses for different combination of the initial state ($\vec{m}\uparrow$, $\vec{m}\downarrow$) corresponding to (**a,b**) and (**c,d**) respectively and laser helicities ($\sigma^+$, $\sigma^-$) corresponding to (**a–c**) and (**b–d**) respectively under the influence of the IFE ($F = 0.55$ J/m$^2$, $B_{MO}^0 = 5$ T ). Each pulse is applied every $\Delta t_L = 1$ ns. The last snapshot shows the final relaxed state after 25 laser pulses. A video of these HD-AOS is provided as Supplementary Material (video S1).

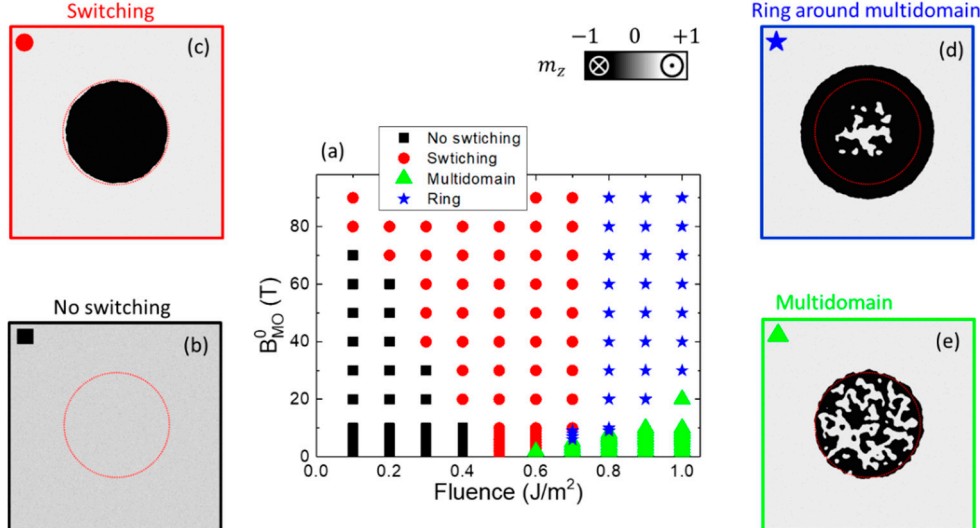

**Figure 3.** (**a**) Phase diagram of final states as a function of the fluence (*F*) and the maximum amplitude of the magneto-optical field ($B_{MO}^0$) starting from a uniform up magnetization ($\vec{m} \uparrow$ ). The results were obtained by applying a train of laser pulses with left-handed helicity ($\sigma^-$). Snapshots (**b**–**e**) represent typical final possible states. The black squares correspond to no-inversion combinations (**b**). The red dots correspond to cases where the local magnetization under the laser beam reverses its initial state (**c**). The green triangles correspond to a multi-domain final state (**d**), and the blue stars depict combinations of where the central area below the laser beam becomes demagnetized with an inverted ring around a multi-domain pattern (**e**).

Four different final states can be obtained depending on *F* and $B_{MO}^0$ (see Figure 3). For low fluences, i.e., $F < 0.5 \, \text{J/m}^2$, electron temperature remains always below $T_C$. For those fluences, the magnetization dynamics only depicts thermal noise fluctuations, and random domains which have transitorily inverted its initial magnetization finally collapse under the action of very low moderate magneto-optical field $\vec{B}_{MO}$ ($B_{MO}^0 < 1 \, \text{T}$). These combinations of ($F, B_{MO}^0$) result in a non-inverted state (black squares in Figure 3a,b). The local switching is achieved if $B_{MO}^0$ is high enough for $F < 0.5 \, \text{J/m}^2$ (red circles in Figure 3a,c). Thermal agitation along with $\vec{B}_{MO}$ produce the inversion of the magnetic state in the central part of the sample, as indicated by red dots in Figure 3a. An example of the final inverted state is shown in Figure 3c. Note that the critical value of the $B_{MO}^0$ decreases gradually as *F* increases.

For $F \geq 0.5 \, \text{J/m}^2$, the electron temperature overcomes $T_C$ in an extended region of the sample, and the ferromagnetic order is lost in the central part of the sample during every laser pulse. After a few fs, the electron temperature decreases under $T_C$ (see Figure 1 as an example of the electron temperature evolution) and magnetic moments are remagnetized under the presence of $\vec{B}_{MO}$, that may last longer than the laser pulse or maintain a high enough value when the sample is below $T_C$. Therefore, the sample reverses its initial state even for moderate $\vec{B}_{MO}$. If the fluence is further increased, the electron temperature also increases, and it takes longer to cool down to $T_C$. In this situation two cases are possible: if the $B_{MO}^0$ is relatively low, the sample ends in a random multi-domain configuration consisting on up and down domains under the laser spot (Figure 3e). For higher $B_{MO}^0$ values, there is a region around the center of the laser spot which presents moderate values of $T_e$ and $B_{MO}^0$. These combinations lead to magnetic inversion in a surrounding ring around the center of the laser spot. However, $T_e$ is still over $T_C$ in the central part when the $\vec{B}_{MO}$ disappears, which promotes the multi-domain central pattern. Consequently, the final state depicts an inverted external ring around the central multi-domain core (Figure 3d).

Real experiments depict similar four final magnetization patterns when the laser fluence is varied [10]. Although $F$ can somehow be experimentally estimated, the local magnitude of $\vec{B}_{MO}$, which is time and space dependent, remains essentially unknown. According to our results, we find transitions passing through non inversion (Figure 3b), single domain central switching (Figure 3c), multi-domain state surrounded by a ring (Figure 3c), and fully demagnetized state (Figure 3e), which reproduces the experimental results as a function of fluence for a given $B_{MO}^0$. Therefore, our model could provide a realistic determination of the magnitude of $\vec{B}_{MO}$ by direct comparison to experimental measurements.

On the other hand, each laser pulse is usually applied every millisecond (ms) in experimental setups. With such a repetition rate (1 pulse per ms), it is possible to obtain images of the magnetic state after several ms. However, due to the limitation of the computing time, it is not possible to simulate such pulse repetition rates. Nevertheless, we also performed simulations changing the time between consecutive laser pulses ($\Delta t_L$) from 0.5 ns to 12 ns. For very fast repetition rates ($\Delta t_L < 5$ ns), the sample is not cooled down to 300 K after each pulse, but after five pulses, there is no more heat accumulation and the substrate extracts all the energy of the following pulses. This small substrate temperature increase does not change substantially the quantitative behavior. The results confirm that the phase diagram (Figure 3a) does not qualitatively depend on the separation between pulses. As expected, the inverted area increases as the separation time between consecutive pulses decreases, and it converges with the number of laser pulses (Figure 4a,b). Although the experiments show results for a very high number of pulses and pulse separation of several ms that cannot be simulated due to computing restrictions, our micromagnetic studies are still valid for comparison.

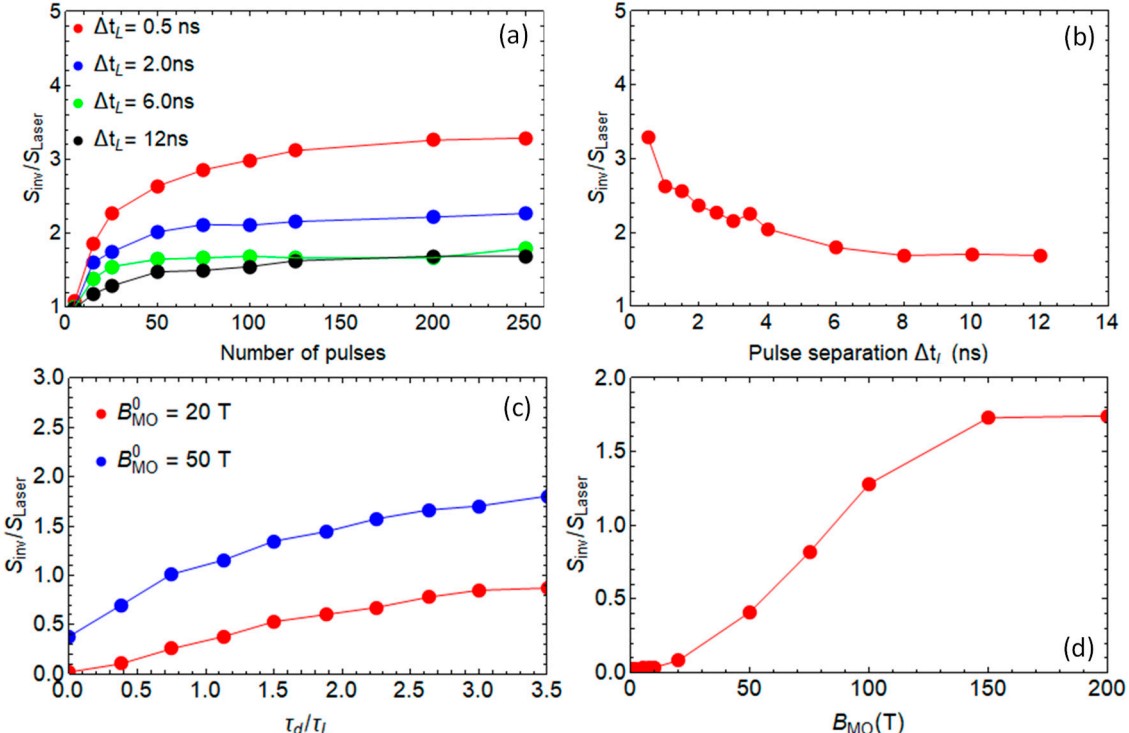

**Figure 4.** (**a**) Inverted area ($S_{inv}$) normalized to the area of the laser spot ($S_{Laser} = \pi r_0^2$ with $r_0 = d_0/2$ being the FWHM laser radius) as a function of the number of pulses for several of the separation time between consecutive pulses ($\Delta t_L$). (**b**) Normalized inverted area ($S_{inv}/S_{Laser}$) as a function of separation time ($\Delta t_L$) between consecutive laser pulses. (**c**) Normalized inverted area ($S_{inv}/S_{Laser}$) as a function of the normalized delay time between the magneto-optical field and the laser duration ($\tau_d/\tau_L$) for two different values of $B_{MO}^0$. (**d**) Normalized inverted area ($S_{inv}/S_{Laser}$) as a function of the $B_{MO}^0$ when the magneto-optical field and the laser duration are the same ($\tau_d = 0$).

Another IFE parameter which cannot be experimentally measured is the temporal delay ($\tau_d$) of the magneto-optical field $\vec{B}_{MO}$ with respect to the laser pulse (see the temporal profile of the laser pulse Equation (1) and the corresponding one of $\vec{B}_{MO}$ in Equation (10)). The theory of the IFE predicts the existence of the transient induced magneto-optic field, but its duration remains essentially unknown. However, the role of such delay can be easily evaluated in our numerical framework. To check the influence of the delay, we carried out simulations changing $\tau_d$, and even assuming that the duration of the $\vec{B}_{MO}$ is the same that the one of the laser pulse ($\tau_d = 0$). As shown in Figure 4c, we verify that it is still possible to achieve HD-AOS in the absence of delay if the $\vec{B}_{MO}$ reaches high enough values.

We can conclude that although the quantitative values of the phase diagram presented in Figure 3 may differ depending on pulse duration ($\tau_L$), number of applied pulses and the delay of $\vec{B}_{MO}$ ($\tau_d$), this IFE scenario predicts the existence of a fluence dependence of the switching and the appearance of four magnetic configurations which are usually found in the experimental literature (see, for example, Figure 1d or Figure 3b of Ref. [10]). Moreover, our numerical model could be used to infer the magnitude of the $\vec{B}_{MO}$ and its possible delay with respect to the laser pulse ($\tau_d$) by direct comparison with high-repetition rate of multi-shot helicity dependent experiments.

### 3.1.2. Helicity-Dependent All Optical Switching with Magnetic Circular Dichroism

In the previous subsection, we have verified that a realistic numerical framework considering the laser heating and the IFE can reproduce the experimental observations of the HD-AOS. Now, we study if the MCD can lead by itself to optical magnetization switching, using the proper combinations of initial magnetization and helicity. After an adequate choice of the fluence and MCD parameters, we plot the snapshots of the magnetization under the same train of pulses as used for the previous IFE analysis. The corresponding results for the four different combinations of the initial magnetic state ($\vec{m} \uparrow$ or $\vec{m} \downarrow$) and the laser pulse helicity ($\sigma^+$ or $\sigma^-$) are shown in Figure 5. Note that in this section the micromagnetic results do not take into account the IFE, and only the MCD is evaluated.

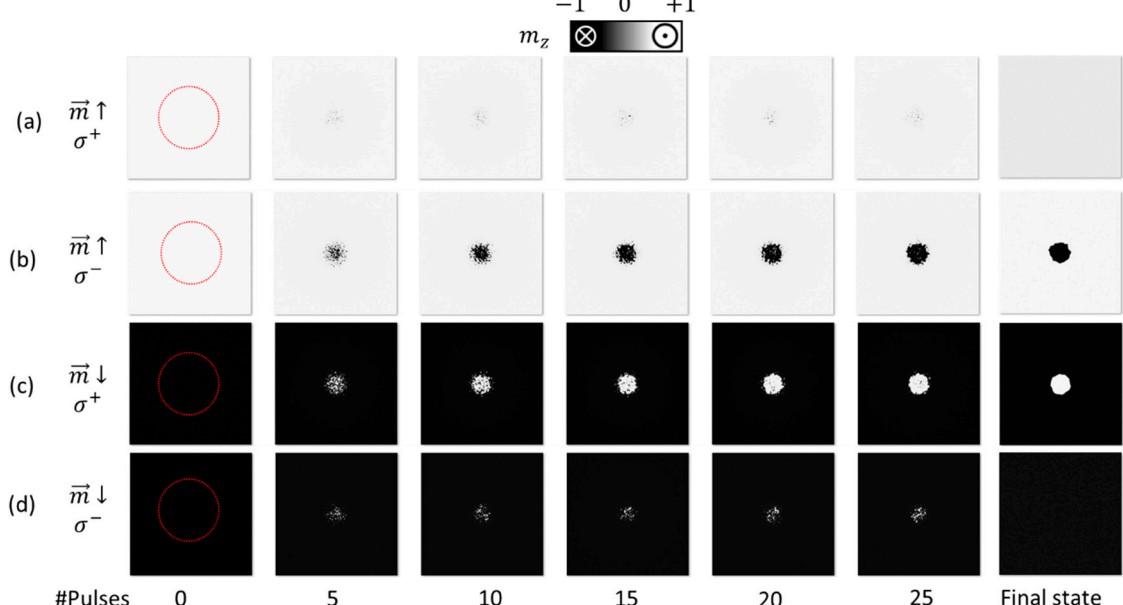

**Figure 5.** Magnetization snapshots after several pulses under the influence of the MCD. Results are shown for right-handed ($\sigma^+$) (**a**) and (**c**) and left-handed ($\sigma^-$) (**b**) and (**d**) circular polarization of the laser beam, and initial magnetic states pointing up ($\vec{m} \uparrow$) (**a**) and (**b**) or down ($\vec{m} \downarrow$) (**c**) and (**d**). The fluence and the MCD absorption rate are $F = 0.5 \text{ J/m}^2$ and MCD = 10% respectively. A video of these HD-AOS is provided as Supplementary Material (video S1).

The results presented in Figure 5 (where the MCD is set on, and the IFE is off) are similar to the ones presented in Figure 2 (where the IFE was set on, and the MCD off). Therefore, the analysis suggests that it is still possible to obtain qualitatively similar final states by IFE and MCD. However, some discrepancies between the final states obtained by IFE and MCD were noticed. If the MCD were the main mechanism of the HD-AOS processes in the evaluated samples, the switching seems to be less clear than in the IFE scenario, with higher noise in the central part under the laser beam. In any case, the MCD seems to be also a valid mechanism that can describe the AOS observations. This is even clearer if we consider the phase diagram where we vary the fluence and the MCD absorption rate. The corresponding MCD phase diagram is shown in Figure 6a.

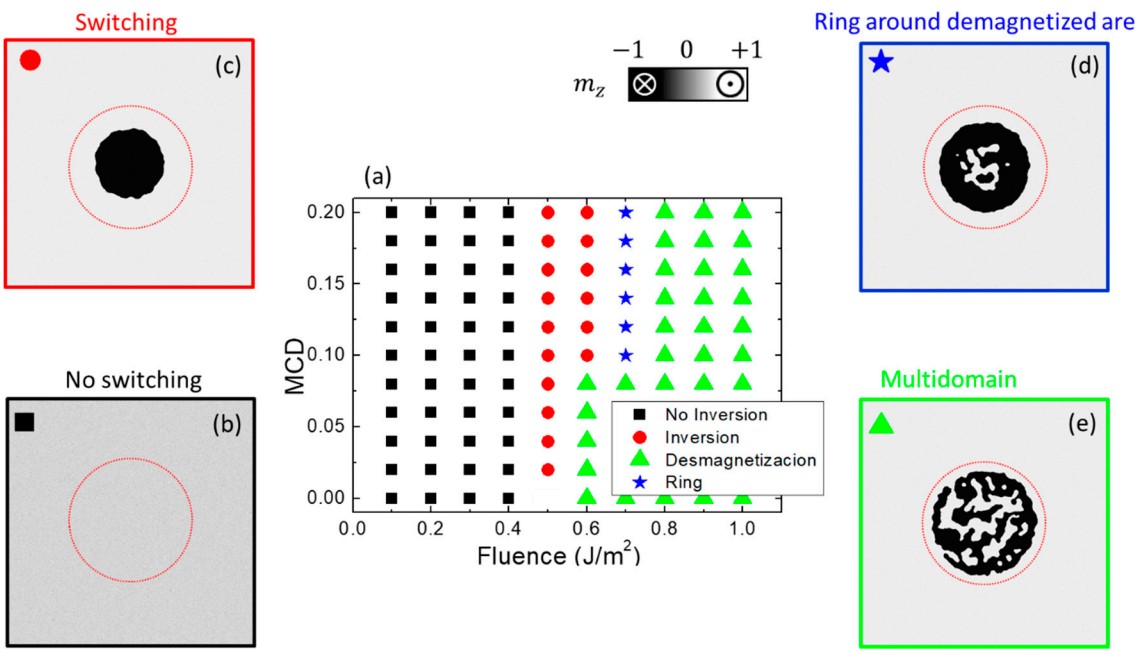

**Figure 6.** (**a**) Phase diagram showing the final states as a function of the fluence and MCD absorption rate starting from a uniform up magnetization ($\vec{m}\uparrow$) and applying a train of 25 laser pulses with left-handed helicity ($\sigma^-$). (**b**–**e**) are representative snapshots of the four possible final states.

Considering the MCD as the only responsible mechanism for the HD-AOS, the final states are similar to those of IFE, with regions that present no inversion (Figure 6b), central inverted domains (Figure 6c) and central multi-domain states, with (Figure 6d) or without (Figure 6e) the surrounding inverted ring. The main difference with respect to the IFE scenario is the small area that corresponds to the central inversion, which appears only over a fluence threshold of about $F = 0.5 \, \text{J/m}^2$ independently on the MCD rate. We confirmed that such abrupt transition at $F = 0.5 \, \text{J/m}^2$ occurs when the central part of the sample overcomes $T_C$ during each laser pulse. In this case, the inversion mechanism is probabilistic: non inverted domains absorb more energy, and as the local temperature decreases again under $T_C$ once the laser pulse is turned off, they may magnetize randomly either up or down. As the regions magnetized down absorb less power from the laser beam, they present less probability of another demagnetization than those with already up magnetization. Additionally, the external part of the sample that remains magnetized up produces a dipolar field that aids to the remagnetization in the down configuration. For higher fluences ($F > 0.5 \, \text{J/m}^2$) the thermal noise increases, and the region where temperature passes through $T_C$ presents a random multi-domain magnetization. The external region, with lower temperature, and therefore smaller thermal noise, presents an inverted region, leading to similar states as those obtained from the IFE scenario. The main differences of the MCD phase diagram (Figure 6a) as compared to the IFE phase diagram (Figure 3) are the reduced region of the phase diagram that leads to clear inversions or central multi-domain states surrounded

by a ring for the MCD case. Moreover, the MCD absorption rate requires values of 10% in order to lead to the four possible configurations. Additionally, it was also verified that the size of the inverted central domain (see Figure 6c) is smaller than the FWHM of the laser beam, and contrary to the IFE case, this central domain does not expand with the number of laser pulses. Taking into account these differences, the IFE seems to be more plausible to reproduce the experimental observations. However, from the phase diagrams, it is still not possible to conclude which mechanism is dominant in typical HD-AOS experiments. In the following section, we submit the IFE and the MCD to another different numerical test in order to check their ability to reproduce experimental observations.

### 3.2. Helicity-Dependent Domain Wall Motion (HD-DWM)

Light–matter interaction has been also observed in high PMA ferromagnetic thin films by performing domain wall (DW) motion experiments. When focusing a circularly polarized train of laser pulses close to a DW, its dynamics can be promoted [11]. In order to ensure that the displacement of the DW is caused by the motion and not by the local switching, the fluence in this kind of experiments must be within the no-switching regime discussed in the previous sections. Therefore, for the following results, the fluence is maintained bellow the switching threshold.

Depending on the laser position with respect to the DW location and on the laser helicity, several experiments have been performed. In some of them, the laser beam is focused in the middle of a DW [7]. The DW moves to the right or to the left depending on the laser helicity. If the DW is an up to down transition (up-down DW) and the laser has right-handed circular polarization ($\sigma^+$), the DW is displaced to the right. The same DW is displaced to the left for left-handed polarization ($\sigma^-$). The direction of the DW motion reverses for a down-up DW configuration [8]. The HD-DWM was also experimentally observed when the laser is slightly focused to the left or to the right with respect to the initial position of the DW [11]. For up-down DW and a right-handed (left-handed) helicity, the DW moves to the right (left) when the laser beam is focused at the right side (left side) of the DW position. In the following subsections, we numerically study the HD-DWM as caused by the IFE (Section 3.2.1) and the MCD (Section 3.2.2).

### 3.2.1. Helicity-Dependent Domain Wall Motion with Inverse Faraday Effect

Our initial configuration contains a down-up DW. In this case, we are not interested in the nucleation of inverted domains so we need to be within the range of $F$ and $B_{MO}$ that does not lead to switching. As an example, Figure 7 shows the final position of the DW for $F = 0.3 \text{ J/m}^2$ and $B_{MO}^0 = 10 \text{ T}$ and for different laser positions and polarizations after 100 pulses. The initial state is also shown for visual comparison. Note that those values of $F$ and $B_{MO}$ are in the no-switching range (see Figure 3a), but close enough to have a significant DW displacement.

Micromagnetc simulations lead to similar results (Figure 7) as the experimental observations of Figure 1 in [11]. Indeed, the HD-DWM is achieved when a magneto-optical field $\vec{B}_{MO}$ pointing in the opposite direction than the local magnetization under the laser beam is produced by the proper helicity (Figure 7). In order to emphasize the importance of the laser helicity (i.e., the $\vec{B}_{MO}$ direction), the effect of laser pulses with linear polarization was also evaluated and shown in the bottom snapshots of Figure 7. When the laser pulse is applied, there is a local temperature increase, and the DW motion is favored towards the hotter region (located at the center of the laser beam) by the reduction of the DW energy [32,38]. In the case of linear polarization (bottom graphs in Figure 7), the DWM caused by thermal gradients is negligible as compared to the one produced by effective field $\vec{B}_{MO}$ due to the circular polarization (see top and central in Figure 7a,b). Therefore, such effective field $\vec{B}_{MO}$ due to the IFE is the main responsible for the DWM for both combinations of the relative laser-DW position and laser helicities (see Figure 7). We conclude that the IFE provides a valid explanation for the HD-DWM observations and that the main cause of the DW displacement originates the presence of the

magneto-optical induced effective field $\vec{B}_{MO}$, and not by the thermal gradients (note that laser pulses with linear polarization do not produce an observable DW displacement).

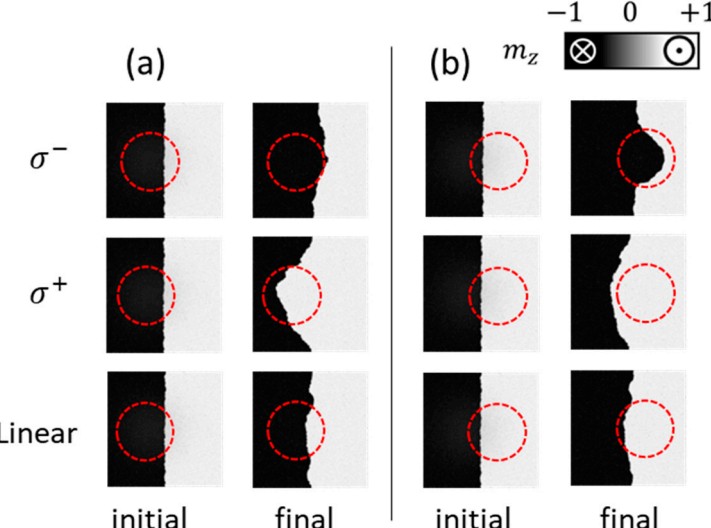

**Figure 7.** Initial and final snapshots of a down-up DW after 100 laser pulses of 200 ps of duration. The magneto-optical effective field due to the IFE has maximum amplitude of $B_{MO,MAX} = 10$ T, and its direction, either up or down, is given by the circular polarization ($\sigma^-$ and $\sigma^+$ for top and central graphs). The fluence is $F = 0.3$ J/m$^2$. The results for linear polarization ($B_{MO,MAX} = 0$) are shown in bottom graphs. The red dotted circles indicate the location of the laser beam. The in-plane size is $1.5 \, \mu m \times 1.5 \, \mu m$, and the center of laser was displaced 192 nm to the left (**a**) or to the right (**b**) of the DW.

### 3.2.2. Helicity-Dependent Domain Wall Motion with Magnetic Circular Dichroism

In order to check whether the MCD is consistent with experimental observations, we also evaluated the HD-DWM processes when instead of the IFE, only the MCD is considered. Similar to previous Section 3.2.1. We worked in the range of MCD absorption rates and fluences which does not lead to the switching but is close to the transition frontier. For instance, for $F = 0.35$ J/m$^2$ and MCD = 10%, a similar behavior to the one of the linear polarization presented in the previous section is expected. Nevertheless, the existence of the MCD implies a different rate of absorption of energy for up and down domains and therefore, an increase of the temperature gradient leading to higher forces pushing the DW to the hot region [35]. Figure 8 shows different final magnetic configurations for the two possible circular polarizations and laser positions (see top and central panels in Figure 8a,b for a laser beam at the left and the right side of the DW, respectively). The case of linearly polarized laser pulses is shown for comparison (bottom snapshots in Figure 8). These results indicate that the HD-DWM is negligible for both circular polarizations and relative laser-DW locations. Note that the DW displacement it is comparable with the one obtained by the linear polarization. Although we plotted one combination of fluence and MCD value in Figure 8, none of the different trials in the no-switching region resulted in appreciable HD-DWM. Although we cannot completely exclude some influence of the MCD in real experiments, these results (Figure 7 for IFE and Figure 8 for MCD) show that the IFE is the dominant phenomena involved HD-DWM observations. A video with some of the results already included in Figures 7 and 8, and showing the DW dynamics in the IFE and MCD scenarios for two laser spot locations and helicities is provided in the supplementary material for comparison (video S2).

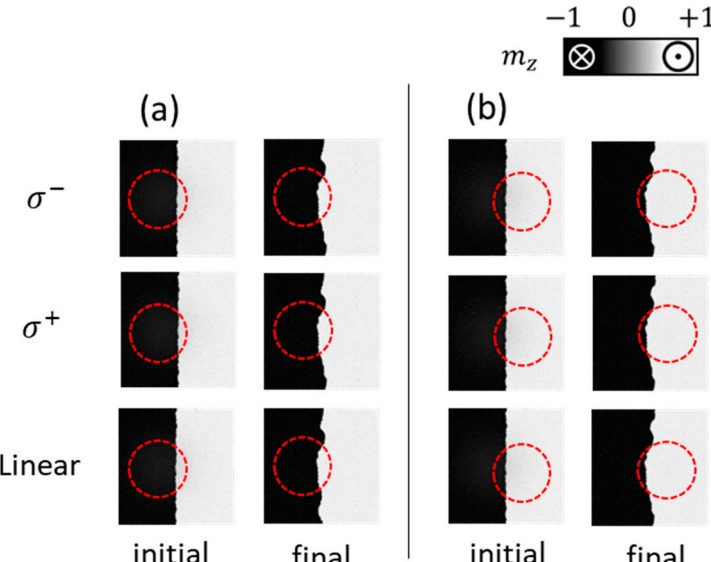

**Figure 8.** Initial and final snapshots of the magnetization of a sample with down and up domains separated by a domain wall. The final magnetic state is obtained after 100 laser pulses of $\tau_L = 200$ ps. A MCD absorption rate of 10% is considered for both circular polarizations (top ($\sigma^-$) and bottom ($\sigma^+$) graphs). The case of linear polarization (MCD = 0) is depicted in bottom graphs. Red dotted circles indicate the location of the laser beam. The in-plane sample size is 1.5 μm × 1.5 μm and the center of laser spot is displaced 192 nm to the left (**a**) or to the right (**b**) of the DW.

## 4. Conclusions

We presented a micromagnetic model based on the LLB equation and the 2TM which also takes into account the IFE and MCD phenomena and allows us to evaluate their real scope in different helicity-dependent all-optical processes. Numerical simulations suggest that helicity-dependent all-optical switching (HD-AOS) in ferromagnetic systems with high PMA can be qualitatively consistent with both IFE and MCD scenarios, which both indicate that the final magnetic state depends on the laser helicity and the initial magnetic state. However, some discrepancies between IFE and MCD results were evidenced. In the IFE scenario, the magnitude of the magneto-optical field needed to promote a single reversed domain decreases as the laser fluence increases. On the contrary, there is a minimum fluence to promote the switching which is independent on the energy absorption rate in the MCD scenario. We also detected a size expansion of the reversed domain with the number laser pulses in the IFE scenario. This expansion is not present when the HD-AOS processes are evaluated in the framework of the MCD scenario. Although the final magnetic states presented in the MCD phase diagram seem to be similar to the ones in the IFE diagram, the numerically required MCD absorption rates (MCD~10%) appears to be significantly higher than the ones expected from experiments (MCD~1%).

The role of the IFE and the MCD has been also evaluated to describe the helicity-dependent DW motion (HD-DWM). In this case, only IFE results are consistent with experimental observations, where the DW is only displaced from its initial location for the correct combination of the relative laser spot location and helicity. On the contrary, the MCD hardly affects the DW position for any of the two circular polarizations of the laser pulses. Indeed, MCD results for both laser helicities (circular polarization) are similar to the ones obtained for linear polarization, where the only effect of the laser pulse is the generation of thermal gradients, which were not sufficient to promote any DW motion in the present analysis. Based on these results, we conclude that the IFE has a dominant role over the MCD to describe the helicity-dependent all-optical manipulation of magnetization in ferromagnetic thin films with perpendicular anisotropy. We believe that our methods can be useful in the next future when estimating the value of some parameters, as the ones related with inverse Faraday effect, which remain essentially unknown nowadays.

**Supplementary Materials:** The following are available online at http://www.mdpi.com/2076-3417/10/4/1307/s1, **Video S1**: Comparison of temporal magnetization evolution of the HD-AOS for the IFE and the MCD scenarios under 25 laser pulses. Up and down initial states and right-handed and left-handed circular polarization are evaluated in both cases. **Video S2**: Comparison of the HD-DWM for the IFE and the MCD scenarios. Right-handed and left-handed circular polarization, and two different locations of the laser spot (either at the left side or the right side of the down-up DW) are presented for both IFE and MCD simulations.

**Author Contributions:** V.R. and E.M. proposed and coordinated the research project. V.R. developed the micromagnetic code. R.G., M.A.H., M.Z., and F.G.-S. performed the simulations. V.R. and E.M. wrote the manuscript. All authors have read and agreed to the published version of the manuscript.

**Funding:** This research was partially funded by Project No. MAT2017-87072-C4-1-P from Ministerio de Economía y Competitividad of the Spanish Government, Project No. SA299P18 from the Consejería de Educación of Junta de Castilla y León and Project MagnEFi, Grant Agreement 860060 (H2020-MSCA-ITN-2019) funded by the European Commission.

**Conflicts of Interest:** The authors declare no conflict of interest.

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
