# Peer review of "Micromagnetic Modeling of All Optical Switching of Ferromagnetic Thin Films: The Role of Inverse Faraday Effect and Magnetic Circular Dichroism"

_applsci, doi:10.3390/app10041307_

Round 1

Reviewer 1 Report

In their manuscript, Raposo et al. present a numerical micromagnetic modeling of the helicity-dependent all-optical switching in a ferromagnetic Pt/Co bilayer. To investigate the role of both the inverse Faraday effect (IFE) and the magnetic circular dichroism (MCD) in the all-optical switching process, the authors coupled the two temperature model (2TM) with the magnetization dynamics governed by the stochastic Laudau-Lifshitz-Bloch (LLB) equation. They showed that both the IFE and the MCD may lead to the all-optical switching. Moreover, they demonstrated that the IFE is more efficient for the domain wall motion and reproduces more accurately the all-optical switching for different values of the laser beam’s fluence.  

Although the micromagnetic formalism of the all-optical switching based on coupling the 2TM with the LLB equation has already been previously reported for ferromagnetic granular FePt in Ref. [11] (Mendil et al., Sci. Rep. 4, 3980 (2014)), the manuscript presents to some extent new results by including the MCD coefficient in the 2TM equation and by simulating the helicity-dependent domain wall motion process with the IFE and the MCD. Nevertheless, my doubts concern the numerical results obtained with the MCD regarding both the helicity-dependent all-optical switching and domain wall motion, which questions the accuracy of the inclusion of the MCD contribution in the micromagnetic model. My recommendation is that this manuscript can be considered for publication in Applied Sciences, provided that it is appropriately revised following the comments below.

(1) In section 3.1.1, the authors studied the helicity-dependent all-optical switching obtained with the IFE. What is the minimum lifetime of the IFE required to obtain the all-optical switching ? Is it comparable with the one reported for ferromagnets using the microscopic three temperature model (M3TM) (see Ref. [24]: Cornelissen et al., Appl. Phys. Lett. 108, 142405 (2016)) ?

(2) In section 3.1.2, the authors showed that the MCD absorption requires values of 10% in order to lead to the possible configurations for the all-optical switching. However, a MCD absorption rate of 10% seems unreasonable and is one order of magnitude larger than the one reported in the literature from both experimental (for ferrimagnetic GdFeCo alloys see Ref. [16]: Khorsand et al., PRL 108, 12705 (2012); for ferromagnetic Co/Pt multilayers see Ref. [8]: Medapalli et al., PRB 96, 224421 (2017)) and theoretical (for ferromagnets see Ref.: Gorchon et al., PRB 94, 020409R (2016)) viewpoints. How do the authors explain this one order of magnitude discrepancy in the MCD absorption rate ?

(3) In section 3.2.2, the authors demonstrated that the helicity-dependent domain wall motion induced by a MCD absorption rate of 10% remains very weak and negligible compared to the one obtained with the IFE. Nevertheless, Quessab et al. (see Ref. [10]: Quessab et al., PRB 97, 054419 (2018)) calculated an effective magnetic field of 7 mT for a MCD absorption rate of 2% and showed that it is sufficient to achieve a clear helicity-dependent domain wall motion in Co/Pt multilayers. Hence, a MCD absorption rate of 10% would lead to an even higher effective magnetic field, and thus to helicity-dependent domain wall motion. The authors should explain within their manuscript the discrepancy between their findings and the ones reported in Ref. [10].

Reviewer 2 Report

I read the manuscript entitled “Micromagnetic modeling of all optical switching of ferromagnetic thin films: the role of inverse Faraday effect and magnetic circular dichroism”. I found that the manuscript was well written, it was clear and easy to follow the discussion of the results. The authors report about the role of the inverse Faraday effect (IFE) and magnetic circular dichroism (MCD) in the helicity-dependent all-optical switching (HD-AOS) of ferromagnetic thin films by carrying out micromagnetic simulations. The authors found that the mechanism based on the IFE was more likely to explain the experimental observations.

It is true that the mechanism of AOS in ferromagnets that requires several laser pulses is still largely unknown and actively debated in the community with the IFE (athermal mechanism) and MCD (purely thermal mechanism) are the two most discussed possible origins of the phenomenon. Though, many theoretical studies have already been published in the field [Scientific Reports, 6-30522 (2016); Scientific Reports, 7-4114 (2017); Phys. Rev. Applied, 6 054004 (2016); Phys. Rev. B, 94 184406 (2016)] that addressed this question, which casts doubt upon the originality of this work. Yet, the authors took a different theoretical approach than the previously reported studies, i.e. taking into account the laser heating effects into realistic micromagnetic simulations based on the LLB equation, which requires heavy numerical calculation time. However, the authors should more deeply discuss the advantages of their theoretical approach compared to other similar published work and to what extent their approach gives more insight into HD-AOS. Finally, the authors were able to theoretically reproduce the recently reported helicity-dependent all-optical domain wall motion (HD-AODWM) that plays an important role in the mechanism of HD-AOS. To my knowledge, this is the first theoretical study reproducing this experimental result. Therefore, this manuscript is of great interest and would give insight into the mechanism of HD-AODWM. However, a better discussion of the parameters used for the IFE and MCD would be required to validate their numerical calculations. As a result, I would suggest publishing this manuscript after minor revisions. Below are my comments. A better referencing of previously published work is strongly recommended as well. (see attached document).
